

Volatility of mixed atmospheric Humic-like Substances and ammonium sulfate particles
Wei Nie[1,2,3,8*], Juan Hong[3], Silja A. K. Häme[3], Aijun Ding[1,2,8*], Yugen Li[4], Chao Yan[3], Liqing Hao[5],
Jyri Mikkilä[3], Longfei Zheng[1,2,8], Yuning Xie[1,2,8], Caijun Zhu[1,2,8], Zheng Xu[1,2,8], Xuguang Chi[1,2,8], Xin
Huang[1,2,8], Yang Zhou[6,7], Peng Lin[6,a], Annele Virtanen[5], Douglas R. Worsnop[3], Markku Kulmala[3],
Mikael Ehn[3], Jianzhen Yu[6], Veli-Matti Kerminen[3] and Tuukka Petäjä[3,1]
[1] Joint International Research Laboratory of Atmospheric and Earth System Sciences, Nanjing University,
Nanjing, China
[2] Institute for Climate and Global Change Research & School of Atmospheric Sciences, Nanjing University,
Nanjing, 210023, China
[3] Division of Atmospheric Sciences, Department of Physics, University of Helsinki, Helsinki, Finland
[4] Division of Environment, Hong Kong University of Science and Technology, Clear Water Bay, Kowloon,
Hong Kong, China
[5] Department of Applied Physics, University of Eastern Finland, Kuopio 70211, Finland
[6] Department of Chemistry, Hong Kong University of Science & Technology, Clear Water Bay, Kowloon, Hong
Kong, China
[7] Key Laboratory of Physical Oceanography, College of Oceanic and Atmospheric Sciences, Ocean University
of China, Qingdao 266100, China
[8] Collaborative Innovation Center of Climate Change, Jiangsu province, China
[a] now at: Environmental Molecular Sciences Laboratory, Pacific Northwest National Laboratory, Richland, WA
20    99532

*Correspondence to: A. J. Ding (dingaj@nju.edu.cn) and W. Nie (niewei@nju.edu.cn)

Abstract
The volatility of organic aerosols remains poorly understood due to the complexity of speciation and
multi-phase processes. In this study, we extracted HUmic-LIke Substances (HULIS) from four
atmospheric aerosol samples collected at the SORPES station in Nanjing, eastern China, and
investigated the volatility behavior of particles at different sizes using a Volatility Tandem Differential





Mobility Analyzer (VTDMA). In spite of the large differences in particle mass concentrations, the
extracted HULIS from the four samples all revealed very high oxidation states (O : C > 0.95),
indicating secondary formation as the major source of HULIS in Yangtze River Delta (YRD). An
overall low volatility was identified for the HULIS samples, with the volume fraction remaining (VFR)
higher than 55% for all the re-generated HULIS particles at the temperature of 280 °C. A kinetic mass
transfer model was applied to the thermodenuder (TD) data to interpret the observed evaporation
pattern of HULIS, and to derive the mass fractions of semi-volatile (SVOC), low-volatility (LVOC)
and extremely low-volatility components (ELVOC). The results showed that LVOC and ELVOC
dominated (more than 80%) the total volume of HULIS. Atomizing processes led to a size dependent
evaporation of regenerated HULIS particles, and resulted in more ELVOCs in smaller particles. In
order to understand the role of interaction between inorganic salts and atmospheric organic mixtures in
the volatility of an organic aerosol, the evaporation of mixed samples of ammonium sulfate (AS) and
HULIS was measured. The results showed a significant but nonlinear influence of ammonium sulfate
on the volatility of HULIS. The estimated fraction of ELVOCs in the organic part of largest particles
(145 nm) increased from 26% in pure HULIS samples to 93% in 1:3 (mass ratio of HULIS:AS) mixed
samples, to 45% in 2:2 mixed samples, and to 70% in 3:1 mixed samples, suggesting that the
interaction tends to decrease the volatility of atmospheric organic molecular once condensing on
ammonium sulfate containing aerosols. Our results demonstrate that HULIS are important low volatile,
or even extremely low volatile, compounds in the organic aerosol phase. As important formation
pathways of atmospheric HULIS, multi-phase processes, including oxidation, oligomerization,
polymerization and interaction with inorganic salts, are indicated to be important sources of low
volatile and extremely low volatility species of organic aerosols.
1.  Introduction
Atmospheric organic aerosol (OA) comprises 20-90% of the total submicron aerosol mass depending
on location (Kanakidou et al., 2005; Zhang et al., 2007; Jimenez et al., 2009), and play a critical role in
air quality and global climate change. Given the large variety of organic species, OA is typically
grouped in different ways according to its sources and physicochemical properties. These include the
classifications based on aerosol optical properties (brown carbon and non-light absorption OA),
formation pathways (primary (POA) and secondary (SOA) organic aerosol) and solubility (water
soluble OA (WSOA) and water insoluble OA (WISOA)). HUmic-LIke Substances (HULIS), according





to their operational definition, are the hydrophobic part of WSOA, and contribute to more than half of
the WSOA (e.g. Krivácsy et al., 2008). Secondary formation (Lin et al., 2010b) and primary emission
from biomass burning (Lukács et al., 2007; Lin et al., 2010a) have been identified as the two major
sources of atmospheric HULIS. Because they are abundantly present, water-soluble, light-absorbing
and surface-active, HULIS in atmospheric particles have been demonstrated to play important roles in
several processes, including cloud droplet formation, light abortion and heterogeneous redox activities
(Kiss et al., 2005; Graber and Rudich, 2006; Hoffer et al., 2006; Lukács et al., 2007; Lin and Yu, 2011;
Verma et al., 2012; Kristensen et al., 2012).
Volatility of atmospheric organic compounds is one of their key physical properties determining their
partitioning between the gas and aerosol phases, thereby strongly influencing their lifetimes and
concentrations. Atmospheric OA can be divided into semi-volatile organic compounds (SVOC), low
volatility organic compounds (LVOC) and extremely low volatility organic compounds (ELVOC)
(Donahue et al., 2012; Murphy et al., 2014). LVOC and ELVOC are predominantly in the aerosol
phase and contribute largely to the new particle formation and growth (Ehn et al., 2014), while SVOC
have considerable mass fractions in both phases and usually dominate the mass concentration of OA.
As far as we know, volatility studies on OA have mostly focused on laboratory-generated organic
particles or ambient particles (Kroll and Seinfeld, 2008; Bilde et al., 2015). Laboratory-generated
organic particles contain only a small fraction of compounds present in atmospheric OA, whereas
ambient particles are usually complex mixtures of thousands of organic and several inorganic
compounds. One way to interlink laboratory and ambient studies, and to understand the volatility of
ambient OA systematically, might be to isolate some classes of OA from ambient particles before
investigating their volatility separately. As an important sub-group of organic aerosols in the real
ambient aerosols, the physicochemical properties of HULIS have been studied widely, including their
mass concentrations (Lin et al., 2010b), chemical composition (Lin et al., 2012; Kristensen et al., 2015;
Chen et al., 2016), density (Dinar et al., 2006) and hygroscopicity (Wex et al., 2007; Kristensen et al.,
2014). However, to the best of our knowledge, the volatility of atmospheric HULIS has never been
reported so far.
In the ambient aerosol, organic aerosol (OA, including HULIS) mostly co-exist with inorganic
compounds, such as ammonium sulfate. The volatility of OA has been demonstrated to be affected by
aerosol-phase reactions when mixed with inorganic compounds (Bilde et al., 2015). The most typical



examples of these are interactions between particulate inorganic salts with organic acids to form
organic salts, which evidently can enhance the partitioning of organic acids onto the aerosol phase
(Zardini et al., 2010; Laskin et al., 2012; Häkkinen et al., 2014; Yli-Juuti et al., 2013;). Recent studies
have reported that the saturation vapor pressure ($p_{sat}$) of ammonium oxalate is significantly lower than
that of pure oxalic acid, with $p_{sat}$ being around $10^{-6}$ Pa for ammonium oxalate (Ortiz-Montalvo et al.,
2014; Paciga et al., 2014). However, this has not shown to be the case for adipic acid vs. ammonium
adipate, indicating that not all dicarboxylic acids react with ammonium to form low-volatility organic
salts (Paciga et al., 2014). Given that HULIS contain acidic species (Paglione et al., 2014; Chen et al.,
2016), their interaction with inorganic salts would plausibly influence their volatility.
In this study, HULIS were extracted from PM$_{2.5}$ filter samples collected at the SORPES station (Station
for observing Regional Processes of the Earth System) in western Yangtze River delta (YRD) during
the winter of 2014 to 2015. A Volatility-Hygroscopicity Tandem Differential Mobility Analyzer
(VHTDMA) was then used to measure the volatility properties of extracted HULIS and their mixtures
with ammonium sulfate. A kinetic mass transfer model was deployed to re-build the measured
thermograms, and to separate the mixture into three volatility fractions having an extremely low
volatility, low volatility and semi-volatility. Our main goals were (1) to characterize the volatility of
size-dependent, re-generated HULIS particles and to get insight into the relationship between
atmospheric HULIS and ELVOC, and (2) to understand how the interaction between HULIS and
inorganic salts affect their volatilities.
2.  Methods
2.1 Sample collection and HULIS extraction
The SORPES station is located on the top of a hill in the Xianlin campus of Nanjing University, which
is about 20 km east from the downtown Nanjing and can be regarded as a regional background site of
Yangtze River delta (YRD) (Ding et al., 2013;Ding et al., 2016). 24-hour PM$_{2.5}$ samples were collected
on quartz filters using a middle-volume PM$_{2.5}$ sampler during the winter of 2014 to 2015. HULIS were
extracted from four aerosol samples for the following volatility measurements.
Water-soluble inorganic ions, organic carbon (OC) and elemental carbon (EC) were measured online
using a Monitor for Aerosols and Gases in Air (MARGA) and a sunset OC/EC analyzer during the
sampling periods. WSOC were extracted from portions of the sampled filters using sonication in



ultrapure water with the ratio of 1 mL water per 1 cm$^2$ filter. Insoluble materials were removed by
filtering the extracts with a 0.45 μm Teflon filter (Millipore, Billerica, MA, USA). A TOC analyzer
with a non-dispersive infrared (NDIR) detector (Shimadzu TOC-VCPH, Japan) was used to determine
WSOC concentrations. The aerosol water extracts were then acidified to pH = 2 by HCl and  loaded
onto a SPE cartridge (Oasis HLB, 30 μm, 60 mg / cartridge, Waters, USA) to isolate the HULIS
following the procedure described in Lin et al (2010b). Most of the inorganic ions, low-molecular-
weight organic acids and sugars were removed, with HULIS retaining on the SPE cartridge. A total 20
ml of methanol was then used to elute the HULIS. The eluate was evaporated to dryness under a gentle
stream of nitrogen gas. A part of the HULIS eluate was re-dissolved in 1.0 mL water to be quantified
with an evaporative light scattering detector (ELSD).
2.2. Volatility measurements by VTDMA measurements
The evaporation behavior of HULIS and their mixtures with AS was measured using a Volatility
Tandem Mobility Analyzer, which is part of a Volatility-Hygroscopicity Tandem Differential Mobility
An- alyzer (VH-TDMA) system (Hong et al., 2014). During the measurements, the hygroscopicity
mode was deactivated, so that only the volatility mode of this instrument was functioning. Briefly,
aerosol particles were generated by atomizing aqueous solutions consisting of HULIS and their
mixtures with AS by using an atomizer (TOPAS, ATM 220). Then, a monodisperse aerosol with
particle sizes of 30, 60, 100 and 145 nm were selected by a Hauke-type Differential Mobility Analyzer
(DMA, Winklmayr et al., 1991). The monodisperse aerosol flow was then heated by a thermodenuder
at a certain temperature, after which the number size distribution of the particles remaining was
determined by a second DMA and a condensation particle counter (CPC, TSI 3010). The
thermodenuder was a 50-cm-long stainless steel tube with an average residence time of around 5 s.
The VTDMA measures the shrinkage of the particle diameter after heating particles of some selected
initial size at different temperatures. Conventionally, the volume fraction remaining (VFR), i.e. the
faction of aerosol mass left after heating particles of diameter $D_{\mathrm{p}}$, is used to describe the evaporation
quantitatively. $D_{\mathrm{p}}$ ($T_{room}$) is the initial particle diameter at room temperature. $D_{\mathrm{p}}$ ($T$) is the particle
diameter after passing through the thermodenuder at the temperature $T$.
The VFR can be defined as:



$$\text{VFR}(D_\text{p}) = \frac{D_\text{p}{}^3(T)}{D_\text{p}{}^3(T_{room})}$$    (1)
2.3. Kinetic mass transfer model
A kinetic mass transfer model (Riipinen et al., 2010) was applied to help interpreting the HULIS
evaporation data. The size distribution, chemical composition and physicochemical properties of the re-
generated HULIS particles, as well as the residence time of the particles traveling through the
thermodenuder, were predefined in the model. As an output, the model provided the particle mass
change as a function of the residence time, which can either increase or decrease depending on the
particle composition, volatility of compounds and concentrations of surrounding vapors. With the aim
to reproduce the observed evaporation pattern of HULIS particles measured by the VTDMA, the model
applied an optimization procedure to minimize the difference between the measured and modeled
evaporation curves of the HULIS particles.
In the model, particles were assumed to consist of compounds that can be grouped into three volatility
bins: semi-volatile, low-volatility and extremely low-volatility components. These three "bins" were
quantified by assuming that they had fixed volatilities with $p_\text{sat}$ (298 K) = [$10^{-3}$ $10^{-6}$ $10^{-9}$] Pa. Modeling
was performed for each experiment / sample separately, with 4 samples and 4 different initial particle
sizes ($D_\text{p}$ = 30, 60, 100 and 145 nm), leading to 16 different model runs, each providing information on
how much semi-volatile, low-volatile and extremely low-volatility matter ($X_\text{i}$) was present in the
investigated particles. The initial particle size refers to the particle diameter prior to heating. The values
for $p_\text{sat}$ (298 K) and $\Delta H_\text{vap}$ (see Table 1 and text above) were selected by doing a preliminary test model
runs. With $\Delta H_\text{vap}$ of around [40 40 40] kJ mol$^{-1}$ and $p_\text{sat}$ (298 K) of [$10^{-3}$ $10^{-6}$ $10^{-9}$] Pa the model was
best able to reproduce the observed evaporation curves of the HULIS aerosol. Such low vaporization
enthalpies (referred often as effective vaporization enthalpies) for aerosol mixtures, for example for
SOA from α-pinene oxidation, have been reported also in previous studies (Häkkinen et al.,
2014;Donahue et al., 2005;Offenberg et al., 2006;Riipinen et al., 2010). The molecular weight and
density of HULIS were assumed to be 280 g mol$^{-1}$ (Kiss et al., 2003;Lin et al., 2012) and 1.55 kg m$^{-3}$
(Dinar et al., 2006), respectively.
Volatility information, specifically described as the saturation vapor pressure and vaporization enthalpy
here, of ammonium sulfate was determined by interpreting the evaporation behavior of laboratory-
generated AS particles using the kinetic evaporation model. By setting the saturation vapor pressures





and enthalpy of vaporization of AS as fitting parameters, the optimum solution was obtained by
minimizing the difference between the measured and model-interpreted thermograms of AS particles.
Hence, $p_{sat}$ (298 K) of $1.9 \cdot 10^{-8}$ Pa and $\Delta H_{vap}$ of 97 kJ mol$^{-1}$ for AS were determined and used in the
following analysis.
2.4 AMS measurement for oxygen to carbon ratio
The O : C (Oxygen to carbon) ratios of re-generated HULIS particles were measured using a high-
resolution time-of-flight aerosol mass spectrometer (HR-Tof-AMS, Aerodyne Research Inc., Billerica,
USA). Detailed descriptions of the instrument and data processing can be found in previous
publications (DeCarlo et al., 2006; Canagaratna et al., 2007). The HULIS solution was atomized to
generate poly-dispersed aerosol particles and introduced into AMS. The AMS was operated in V mode
and the data was acquired at 5-min saving intervals. The AMS data were analyzed using standard Tof-
AMS data analysis toolkits (SQUIRREL version 1.57H and PIKA version 1.16H in Igor Pro software
(version 6.22A, WaveMetrics Inc.). For mass calculations, the default relative ionization efficiency
(RIE) values 1.1, 1.2, 1.3 and 1.4 for nitrate, sulfate, chloride and organic were applied, respectively.
The RIE for ammonium was 2.6, determined from the ionization efficiency calibration. In elemental
analysis, the "Improved- Ambient" method was applied to calculate O:C ratios by considering the
CHO$^+$ ion correction (Canagaratna et al., 2015).
3.   Results and discussions
Figure 1 shows the chemical compositions of the four PM$_{2.5}$ samples, and the oxygen to carbon ratio
(O : C) of the extracted HULIS in related samples. The four samples can be classified into two groups
based on their PM$_{2.5}$ concentrations (the sum of all measured chemical compositions), with one group
(samples 1 and 2) having the PM$_{2.5}$ higher than 110 µg/m$^3$ and the other one (samples 3 and 4) having
the PM$_{2.5}$ lower than 40 µg m$^{-3}$. The concentrations of inorganic compounds (sulfate, nitrate and
ammonium) were significantly higher in samples 1 and 2 than in samples 3 and 4. The HULIS
concentrations were also higher in samples 1 and 2 (about 9 µg/m$^3$) than in samples 3 and 4 (about 6
µg/m$^3$). The oxidation states of the HULIS, however, did not show any notable differences, showing
very high values for all the four samples (O:C > 0.95), indicating that the HULIS in YRD could be
mostly secondarily formed even during the relatively clean days. Such high oxidation states suggest
further that the extracted HULIS were very likely highly-oxidized, multifunctional compounds (HOMs)
originating from multi-phase oxidation (Graber and Rudich, 2006).



3.1 Volatility of atmospheric HULIS
The volume fraction remaining (VFR) of the HULIS particles as a function of the heating temperature
obtained from VTDMA is illustrated in Fig. 2. An overall low volatility was identified for the HULIS
particles, with the VFR higher than 55% for the particles of all 4 sizes at the heating temperature of
280 °C and residence time of 5 s. Small differences in the volatility could be observed between the
samples of high mass concentrations and low mass concentrations in that the evaporation of HULIS in
samples 1 and 2 was in general weaker than that in samples 3 and 4. In addition, all the samples started
to evaporate from the very beginning of the heating program (around 20 °C to 25 °C) and the
evaporation curves varied smoothly, suggesting that the HULIS particles were mixtures of compounds
having wide range of saturation vapor pressures.
A kinetic mass transfer model was applied to reproduce the observed evaporation of the HULIS, and to
estimate the mass fractions of semi-volatile (SVOC, $p_{sat}$ (298K) = $10^{-3}$ Pa), low-volatility (LVOC, $p_{sat}$
(298K) = $10^{-6}$ Pa) and extremely low-volatility organic components (ELVOC, $p_{sat}$ (298K) = $10^{-9}$ Pa).
As shown in Fig. 3, the model performed reasonably well in simulating the "pure" HULIS particles
(example for sample 1). Noting that the HULIS mixtures were represented with only three model
compounds of different volatilities, the modeled evaporation curves of the HULIS in all samples
showed a relatively good agreement with the measured evaporation curves for all the four particle sizes.
The shape of the modeled thermograms is not as smooth as that of the measured ones suggesting lower
number of volatilities in simulations compared with in the real samples. The model-simulated
distributions of SVOC, LVOC and ELVOC of each HULIS sample gave indication on the volatility of
HULIS. As shown in Fig. 4, all the HULIS samples consisted of compounds from all the 3 volatility
"bins", further confirming HULIS to be mixtures of compounds with wide range of volatilities. SVOC
was estimated to account for only small proportion (less than 20% of the particle mass) of the HULIS
samples, while LVOC and ELVOC dominated these samples (78% - 97% of the particle mass),
suggesting an overall low volatility of the extracted HULIS. Given that the heating program has the
potential to raise the evaporation of HULIS by decomposing large molecules, the real volatility of
atmospheric HULIS could be even lower than obtained here.
In spite of their overall low values, the volatilities of the HULIS varied between the different samples.
The HULIS extracted from the samples of higher particle mass loadings (samples 1 and 2) had, in
general, lower volatilities than those extracted from the samples of lower particle mass concentrations





(samples 3 and 4). By taking 30 nm particles as an example, sample 2 had the largest mass fraction of
ELVOC, up to 72%, followed by sample 1 (66%) and sample 3 (64%), while sample 4 had the least
amount of ELVOC (58%). Correspondingly, the mass fraction of SVOC in 30 nm particles was the
highest in sample 4 (9%) and the lowest in sample 2 (6%). Several factors, including the molecular
weight, oxidation state and molecular structure of the compounds, as well as their interaction with other
compounds, can influence the volatility of HULIS. Although there is not enough information to support
the final conclusion, we excluded the oxidation state as a key factor here because its variation did not
match the volatility changes of the HULIS samples. As can be seen from Figs.1 and 4, sample 2
showed the lowest volatility but the third highest oxidation state of the four samples. Instead of the
oxidation state, the interaction between HULIS and inorganic species is a more likely candidate for
influencing the observed variation of the HULIS volatility, especially as the lower-volatility samples
(sample 1 and sample 2) had higher concentrations and fractions of inorganic species (Fig. 1).
Within individual HULIS samples, the estimated amount of ELVOC, LVOC and SVOC varied with the
particle size (Fig. 4). The mass fraction of ELVOC was in the range of 58−72% for the smallest
particles (30 nm in diameter) and decreased to the range of 47−60% for the 60 nm, to the range of
35−53% for the 100 nm particles, and to the range of 20−39% for the 145 nm particles. The amount of
LVOC increased correspondingly with an increasing particle size, from 23−33% for the 30 nm particles
to 52−65% for the 145 nm particles. The amount of SVOC slightly increased with an increasing
particle size, on average from 7.5% (30 nm) to 14.5% (145 nm). The most likely explanation for this
behavior is that, due to the Kelvin effect, compounds with higher volatilities are likely to evaporate
more from smaller particles. This result indicates that size-resolved chemical compositions of
laboratory-generated particles from aqueous solutions of mixtures should be examined more carefully
to support their size-dependent physical properties from lab studies.
3.2 Interaction between HULIS and ammonium sulfate
Interactions between inorganic and organic matter have been shown to influence the volatility of the
organic matter. However, recent work has focused on the interaction between one specific organic
compound and some inorganic salt(s). For example, Laskin et al. (2012) observed the formation of
sodium organic salt in a submicron organic acid-NaCl aerosol. Ma et al. (2013) reported that the
formation of sodium oxalate can occur in particles containing oxalic acid and sodium chloride.
Häkkinen et al. (2014) demonstrated that low-volatility material, such as organic salts, were formed





within aerosol mixtures of inorganic compounds with organic acids. Zardini et al. (2010) and Yli-Juuti
et al. (2013b) suggested that interactions between inorganic salts and organic acids in the particle phase
might further enhance the partitioning of organic acids onto the particle phase. Given the complex
nature of organic aerosols in the real atmosphere, large uncertainties will be induced when using
simplified laboratory results for explaining observations in the real atmosphere. In this study, we
investigated the volatility of mixed samples of HULIS and ammonium sulfate in different ratios in
order to get better understand organic-inorganic interactions under atmospherically relevant conditions.
Three samples were prepared by mixing HULIS (extracted from sample 1) and pure ammonium sulfate
(AS) with the mass ratios (HULIS to AS) of 0.25:0.75, 0.5:0.5 and 0.75:0.25 (actually 0.29:0.71,
0.55:0.45 and 0.79:0.21). As shown by Fig. 5, pure ammonium sulfate particles started to evaporate at
100°C, and were almost entirely evaporated at 180 °C, whereas HULIS aerosol started to evaporate at
the very beginning (about 20 °C) and more than 80% of its volume still remained at 180 °C. The
evaporation curves for the three mixed samples (Fig. 6) showed generally slow evaporation rates within
the temperature windows from 20 °C to 100 °C and from 180 °C to 280 °C, and much faster
evaporation rates between 100 °C and 180 °C. Interactions between HULIS and ammonium sulfate
obviously influenced the observed volatility. For example, the VFRs of 0.25:0.75 samples (Fig. 6a) at
the temperature of 180 °C were around 0.4 (varied from 0.397 to 0.428 for different size particles),
which is significantly higher than the calculated VFR ($0.29 \times 0.8 + 0.71 \times 0.06 = 0.275$) by assuming
HULIS and ammonium sulfate independently separated. This indicates that mixing of ammonium
sulfate to a HULIS solution decreases the volatility of the organic group or, alternatively, forms new
compounds of low volatility. For the 0.5:0.5 and 0.75:0.25 samples (Fig. 6b and 6c), the VFRs at
180 °C were around 0.43 (0.395 to 0.460 for different size particles) and 0.64 (0.595 to 0.655), which
are comparable to the calculated VFR (0.467 for the 0.5:0.5 samples and 0.645 for the 0.75:0.25
samples). These results indicate that the role of HULIS-AS interactions in the volatility of their
mixtures is complex and nonlinear.
In order to quantify the volatility changes of HULIS induced by its interaction with ammonium sulfate,
the kinetic mass transfer model was again applied to estimate the mass fractions of SVOC, LVOC and
ELVOC for the HULIS part in the mixed samples. As shown in Fig. 7, the model's performance in
simulating mixed HULIS-AS samples was fairly good, yet poorer than in simulating the "pure" HULIS
sample. The poorest agreement between the simulated and measured evaporation curves was found for



the 1:3 mixed samples (mass ratio of HULIS to AS), indicating relatively high uncertainties in the
calculated mass fractions of compounds with different volatility bins for this mixture. These visible
differences between modeled and measured results indicate that interactions between HULIS and AS
indeed influence their volatility distribution. As can be seen from Fig. 8, the estimated fraction of
ELVOC in the HULIS part of the 0:25:0.75 (Fig. 8b) and 0.75:0.25 (Fig. 8d) samples was much higher
than in the pure HULIS sample (Fig. 8a), while the ELVOC fraction in the 0.5:0.5 sample was
comparable to that in the pure HULIS sample. By taking 30 nm and 145 nm particles as an example,
the corresponding estimated ELVOC fractions were 0.66 and 0.26 in the pure HULIS sample, 1.0 and
0.93 in the 0.25:0.75 sample, 0.53 and 0.45 in the 0.5:0.5 sample, and 0.83 and 0.71 in the 0.75:0.25
sample, respectively. In spite of the possible overestimation of ELVOCs fraction in 1:3 mixed samples,
these results suggest that the interaction between HULIS and ammonium sulfate tend to decrease the
volatility of HULIS, and that this effect is nonlinear.
4.Conclusion and implication
In this study, we analyzed the volatility of atmospheric HULIS extracted from four $PM_{2.5}$ samples
collected at the SORPES station in the western YRD of eastern China, and investigated how the
interactions between HULIS and ammonium sulfate affected the volatility of HULIS aerosol fraction.
Overall, low volatilities and high oxidation states were identified for all the four samples, with VFRs at
280°C being higher than 55 % and O to C ratio being higher than 0.95 for all the re-generated HULIS
particles. A kinetic mass transfer model was deployed to divide the HULIS mixture into SVOC, LVOC
and ELVOC groups. We found that HULIS were dominated by LVOC and ELVOC (more than 80%)
compounds. Given the possible thermo-decomposition of large molecules during the heating program,
an even lower volatility than found here is possible for atmospheric HULIS in eastern China. The
Kelvin effect was supposedly taking place in atomizing the solutions of the HULIS mixtures, which
resulted in a size dependent distribution of the relative fractions of SVOC, LVOC and ELVOC in the
generated particles. The interaction between HULIS and ammonium sulfate was found to decrease the
volatility of the HULIS part in the mixed samples. However, these volatility changes were not linearly
correlated with the mass fractions of ammonium sulfate, indicating a complex interaction between the
HULIS mixture and inorganic salts.
This study demonstrates that HULIS are important low volatility and extremely low volatility
compounds in the aerosol phase, and sheds new light on the connection between atmospheric HULIS




and ELVOCs. In a view of the important sources of HULIS, multi-phase processes, including multi-
phase oxidation, oligomerization, polymerization and interaction with inorganic salts, have the
potential to lower the volatility of organic compounds in the aerosol phase, and to influence their gas-
aerosol partitioning. Multiphase processes could be one of the important reasons that most models tend
to underestimate the formation of SOA.
**Acknowledgements**
This work was funded by National Natural Science Foundation of China (D0512/41675145 and D0510/
41505109), and the National Key Research Program (2016YFC0202002 and 2016YFC0200506).

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





Table 1 Kinetic model input settings for three-component HULIS aerosol.

| Model input parameter | Unit | HULIS |
|---|---|---|
| Molar mass, $M$ | g mol$^{-1}$ | [280 280 280] |
| Density, $\rho$ | kg m$^{-3}$ | [1550 1550 1550] |
| Surface tension, $\sigma$ | N m$^{-1}$ | [0.05 0.05 0.05] |
| Diffusion coefficient, $D$ | $10^{-6}$ m$^2$ s$^{-1}$ | [5 5 5] |
| Parameter for the calculation of $T$-dependence of $D$, $\mu$ | - | [1.75 1.75 1.75] |
| Saturation vapor pressure, $p_{sat}$ (298 K) | Pa | [$10^{-3}$ $10^{-6}$ $10^{-9}$] |
| Saturation vapor concentration, $c_{sat}$ (298 K) | µg m$^{-3}$ | [$10^2$ $10^{-1}$ $10^{-4}$] |
| Enthalpy of vaporization, $\Delta H_{vap}$ | kJ mol$^{-1}$ | [40 40 40] |
| Mass accommodation coefficient, $\alpha_m$ | - | [1 1 1] |
| Activity coefficient, $\gamma$ | - | [1 1 1] |
| Particle initial diameter, $d_p$ | nm | 30, 60, 100, 145 |
| Particle total mass, $m_{p,tot}$ | µg m$^{-3}$ | 1 |
|  |  | **Thermodenuder** |
| Length of the flow tube | m | 0.50(i.d of 6 mm) |
| Residence time | s | 5 |





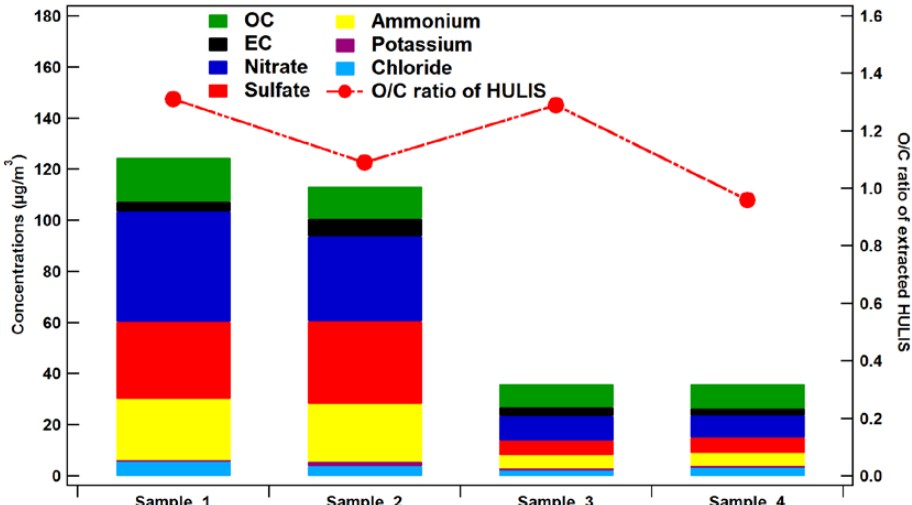

Figure 1 Chemical composition of the four PM$_{2.5}$ samples collected at SORPES station and oxygen
to carbon ratio of extracted HULIS from related samples





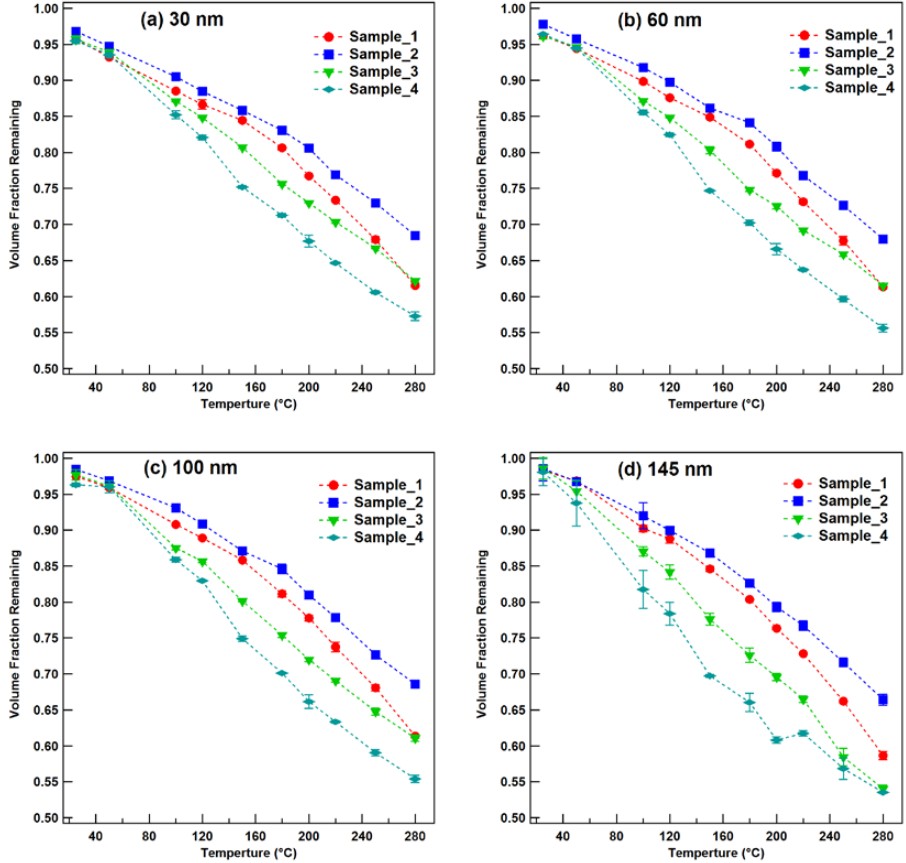

Figure 2 Volume fraction remaining (VFR)as a function of heating temperature for 4 samples at

four different sizes of (a) 30 nm, (b) 60 nm, (c) 100 nm, and (d) 145 nm



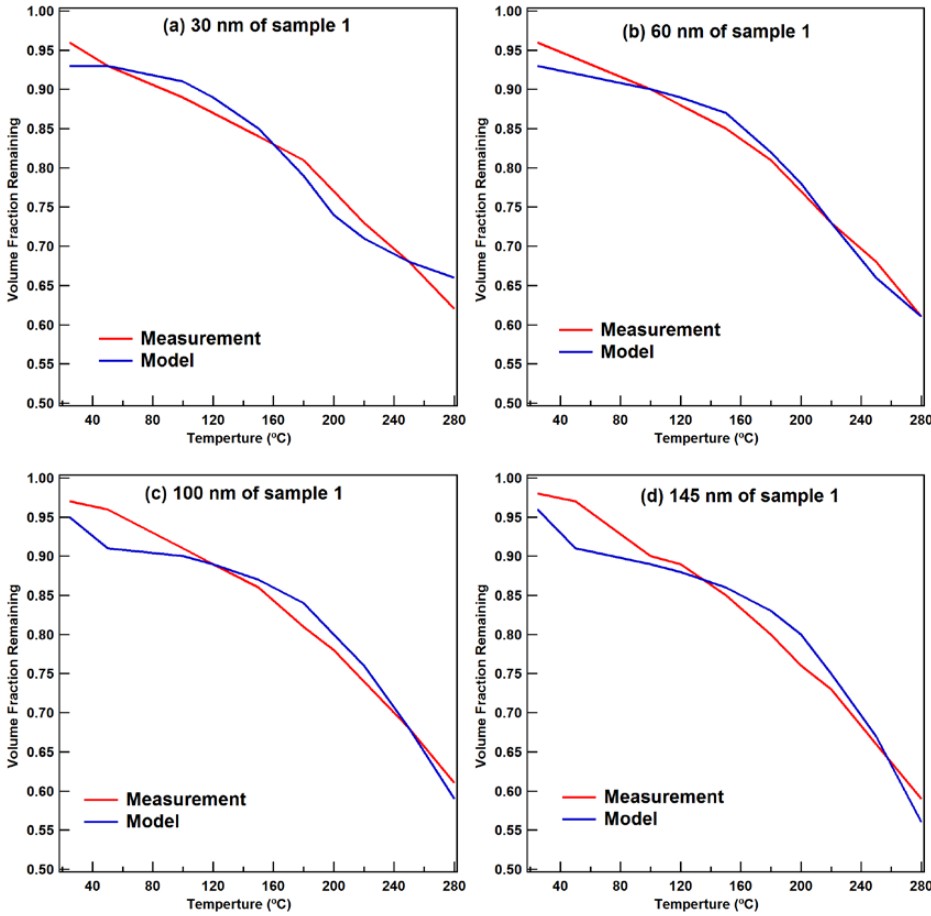

Figure 3 Measured and modeled volume fraction remaining (VFR) as a function of temperature for

HULIS of sample 1 at four different particle sizes of (a) 30 nm, (b) 60 nm, (c) 100 nm and (d) 145 nm





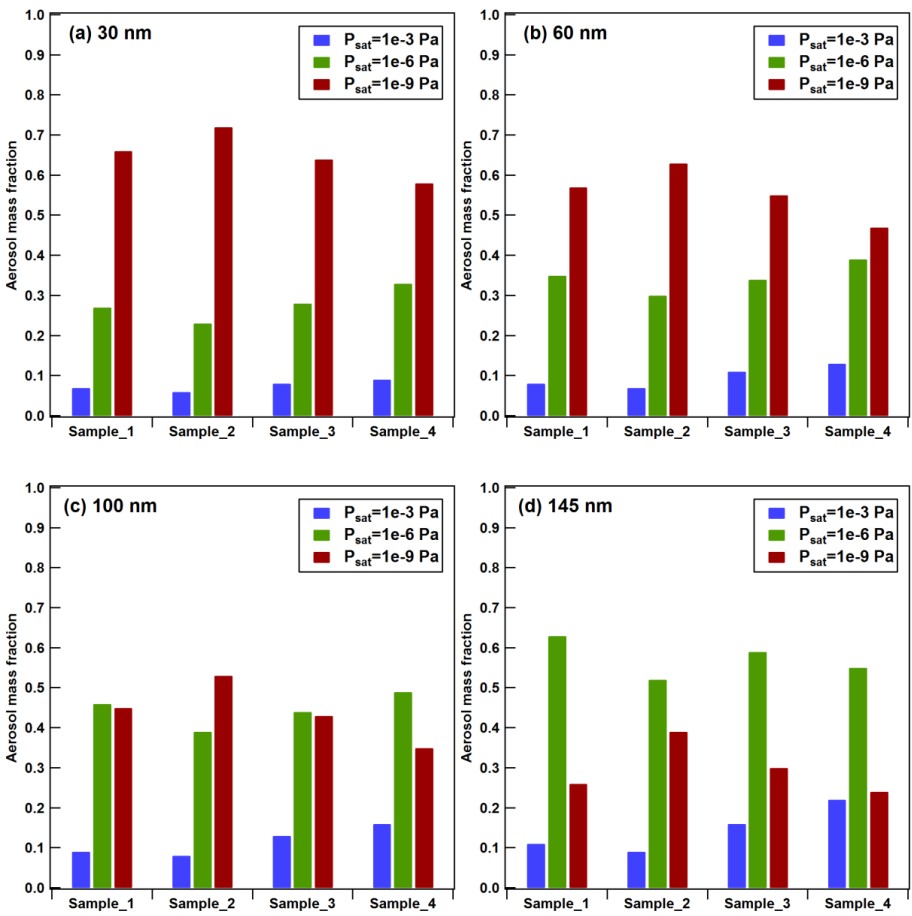

Figure 4 Mass fractions of compounds of SVOC ($p_{sat}=10^{-3}$ Pa), LVOC ($p_{sat}=10^{-6}$ Pa) and ELVOC, ($p_{sat}=10^{-9}$ Pa) in four aerosol samples with different particle sizes of (a) 30 nm, (b) 60 nm, (c) 100 nm, and (d) 145 nm





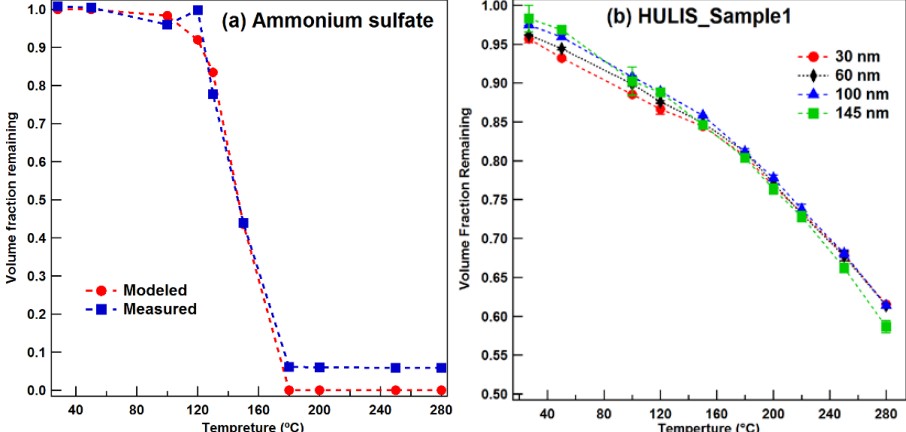

Figure 5 Volume fraction remaining (VFR) as a function of heating temperature for (a) measured and modeled pure ammonium sulfate particles at 100 nm, and (b) HULIS sample 1 at four different sizes of 30 nm, 60 nm, 100 nm, and 145 nm





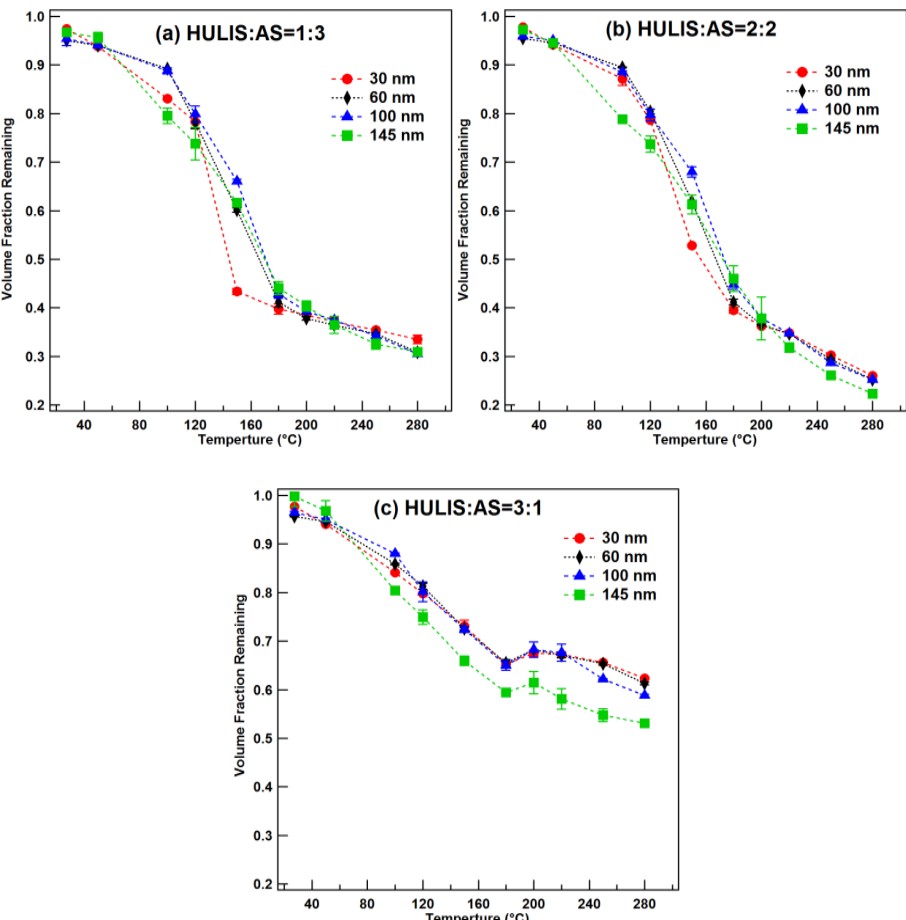

Figure 6 Volume fraction remaining (VFR) as a function of heating temperature for (a) 1:3 HULIS-AS mixed sample, (b) 2:2 HULIS-AS mixed samples, and (c) 3:1 HULIS-AS mixed samples at four different sizes of 30 nm, 60 nm, 100 nm, and 145 nm





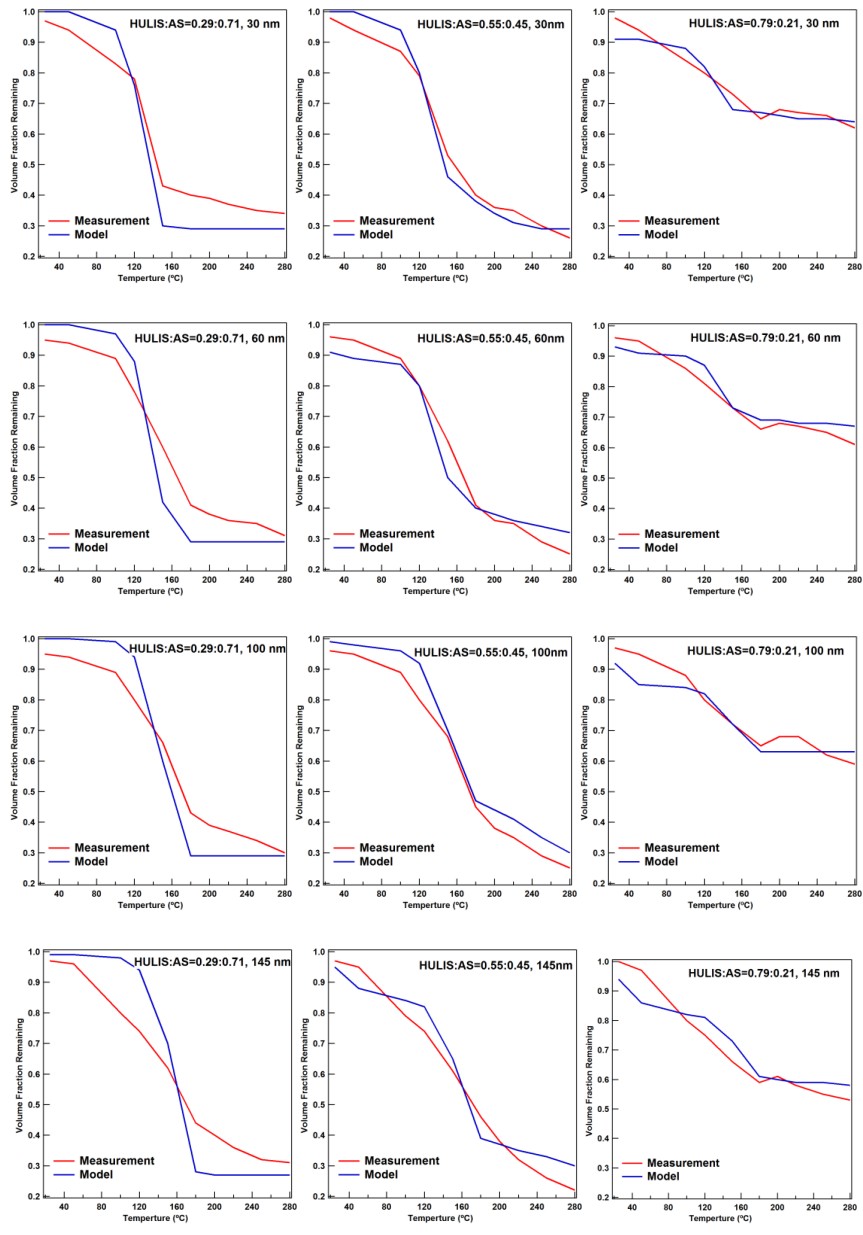

Figure 7 Measured and modeled volume fraction remaining (VFR) as a function of temperature for HULIS-AS mixed samples of 3 different mixing ratios at four different particle sizes of 30 nm, 60 nm, 100 nm and 145 nm



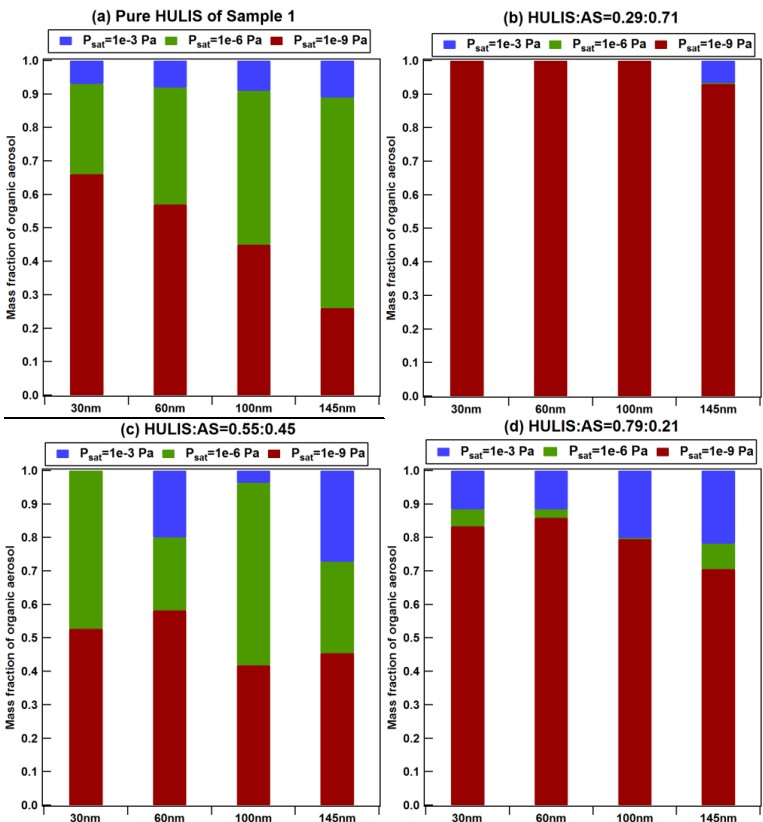

Figure 8 Model-derived mass fractions of organic compounds with different volatilities in four aerosol samples
with different particle sizes of (a) 30 nm, (b) 60 nm, (c) 100 nm, and (d) 145 nm