# Peer review of "Volatility of mixed atmospheric Humic-like Substances and ammonium sulfate particles"

_Atmospheric Chemistry and Physics, 2016_

## Referee Comment (RC1) · Anonymous Referee #3 · 19 Oct 2016

In my opinion the submitted manuscript addresses an interesting scientific issue, i.e., the study on the volatility of HULIS aerosol compounds. The text is understandable for the reader. The objectives are clearly defined and raised conclusions are coherent. However, I would opt to likely consider moving the manuscript first to the ACPD full review process. This is because I have a few remarks as stated below:

**A) Limited number of ambient samples**

The authors made their research based only on 4 filter samples. No justification was provided as to the choice of such a limited number of samples. Certainly, during the full ACPD review phase the authors would be requested to build up their scientific story on the results obtained from the analysis of more ambient samples, say 10 or larger due to the increased complexity of HULIS (e.g., *Environ. Sci. Technol.*, 2016, 50 (4), 1721).

**B) Analytical procedure for the preparation of HULIS extracts**

The authors blindly believe that a whole fraction of HULIS is greatly soluble in the aqueous phase. However, in the light of recent papers dealing with the chemical composition HULIS (e.g., *Environ. Sci. Technol.*, 2016, 50 (4), 1721; *Atmos. Chem.*, 2015, 72, 65), aerosol-derived HUmic LIke Substances represent a complex chemical mixtures, including high-molecular-weight aliphatics (primarily $C_{27}$–$C_{32}$) with small proportions of $-CH_3$, $-OH$, and $C=O$ groups. These are poorly soluble in water, thus the extraction with pure water only may lead to the substantial loss of analyte to be further subjected to the H-TDMA analysis. I am highly surprised that the authors did not take it into account despite it is a basic approach in the analytical atmospheric chemistry: a sample preparation is the most crucial factor. I would suggest broadening a result discussion with providing additional data obtained for samples extracted with less polar solvents, say acetonitrile and water acetonitrile (50/50v). The same could be applied for another solvent couple: methanol and methanol-water (50/50), as recently have been suggested by Lin (*Environ. Sci. Technol.*, 2014, 48(20), 12012). Moreover, I am a bit concern of the selection of SPE method for HULIS water extracts since it may result in a dramatic loss of highly oxidized and water soluble products, such as organosulfates (nitroxyorganosulfates).

---

## Referee Comment (RC2) · Anonymous Referee #4 · 17 Dec 2016

General Comments

Here the authors report results of a laboratory study of the volatility of HULIS extracted from aerosol samples collected at a rural site in eastern China. Samples were atomized, four different sizes were selected with a DMA, and then aerosol was passes through a thermal denuder to measure changes in size with increasing temperature. Extracts were also mixed with ammonium sulfate prior to atomization to investigate the effects of salt-organic interactions on volatility. The volatility profiles were analyzed using a model in which various parameters (heat of vaporization, molecular weight, etc.) were assigned based on previous studies and the aerosol was distributed among three volatility bins (SVOC, LVOC, ELVOC) using the model-measurement comparison. The results of AMS measurements indicate that HULIS is highly oxidized (O/C $\sim$ 1 or greater) and the volatility measurements show that most of the HULIS is low and

extremely low volatility material, consistent with the high degree of oxidation. Small decreases in volatility were also observed when ammonium sulfate was added that indicate chemical interactions between the organic and inorganic materials. The explanations for the general trends observed in the data and model results are reasonable, and overall there are no real surprises. This is a pretty straightforward study, the experimental and modeling components are well done, and the data interpretation is reasonable. The paper is a useful contribution to the literature and is worthy of publication in ACP. I have only a few minor comments that should be addressed.

Specific Comments

1. Line 194–197: What is the fraction of HULIS in the organic component of the samples?

2. Line 199–201: Is auto-oxidation a potential source for these HOMs?

3. What are the effects of assumed model parameters on the interpretation of experimental results? Were sensitivity studies conducted? For example, there is ongoing debate about the appropriate value of the mass accommodation coefficient, which may range from about 1 to 0.001. Couldn't changes in these parameters with organic and inorganic composition be responsible for the observations rather than changes in the SVOC, LVOC, and ELVOC fractions? Some discussion of these issues is needed.

Technical Comments

Line 61: "Abortion" should be "absorption".

Line 128: "An- alyzer" should be "Analyzer".

———————————————————

---

## Referee Comment (RC3) · Anonymous Referee #1 · 24 Dec 2016

This manuscript presents the first study to my knowledge of the volatility of humic-like substances (HULIS) in atmospheric aerosols. The results ultimately link HULIS, which has been observed ubiquitously in atmospheric aerosols using filter collection and extraction, with low and extremely low volatility organic material (ELVOC), which has been observed to be similarly common using AMS and thermal denuder-type techniques. The authors also demonstrate that interactions between HULIS and inorganic salts such as sulfate can greatly decrease the already low volatility of the HULIS. This is a very significant contribution to the field, and should be published after a few issues are addressed.

- Another reviewer commented on the relatively few filter samples that were used in this study. Can the authors make an argument to justify this, for example, were the properties of the HULIS on these filters representative of other samples taken at the

same location?

- My main concern is the extensive chemical processing of the HULIS during isolation before analysis. Is there a way for us to know how this may have impacted the volatility or other properties of the material as compared to the real state in ambient aerosols? A discussion of this point in the manuscript is needed.

- Related to my last point, since the HULIS extraction process removes inorganic ions from the sample, are we to understand that the samples in which salts have been re-introduced are more representative of the true volatility of HULIS in atmospheric aerosols? If so, this should be emphasized.

- Some language editing is necessary in places, for example, lines 42-43

---

## Author Comment (AC1) · 8 Feb 2017

Referee 3

In my opinion the submitted manuscript addresses an interesting scientific issue, i.e., the study on the volatility of HULIS aerosol compounds. The text is understandable for the reader. The objectives are clearly defined and raised conclusions are coherent. However, I would opt to likely consider moving the manuscript first to the ACPD full review process. This is because I have a few remarks as stated below:

**A) Limited number of ambient samples**

The authors made their research based only on 4 filter samples. No justification was provided as to the choice of such a limited number of samples. Certainly, during the full ACPD review phase the authors would be requested to build up their scientific story on the results obtained from the analysis of more ambient samples, say 10 or larger due to the increased complexity of HULIS (e.g., *Environ. Sci. Technol.*, 2016, 50 (4), 1721).

Response: Thanks for the comment, which mainly focused on the representative of our samples.

Firstly, we agree that HULIS are a very complex mixture that contains a large number of compounds with different molecular structures. However, based on our results, their overall volatility of different HULIS samples behaved in quite a similar way (which is reasonable as volatility is mostly controlled by molecule weight and oxidation state). In this work, we actually analyzed 8 samples collected during both winter and summer, and covering a wide range of PM concentration from less than 40 µg/m$^3$ to higher than 150 µg/m$^3$. All these 8 samples showed similar evaporation behavior (as showed in the following figure), with some small differences in details. Therefore, in terms of volatility, we believe that there were no large differences between the collected HULIS samples. We finally selected 4 samples with different PM concentrations to represent the HULIS samples at the SORPES station and made the argument simple and clear that ambient HULIS overall showed a low volatility. We will add some statements on this into the revised manuscript.

Secondly, our sampling site, named SORPES, has been demonstrated to be a regional background site of Yangtze River Delta. As showed in the followed 2$^{nd}$ figure (Ding et al., 2013), SORPES station located in the downwind region of YRD, and is about 20 km east

of the downtown Najing city. Compared to the typical urban measurement station, air masses arriving at this site are more aged and well-mixed. Secondary formed aerosol, e.g. sulfate, nitrate and SOA, was dominated the mass of PM$_{2.5}$. The samples collected at this site were thus believed to be regional representation of Yangtze River Delta (Ding et al., 2013 & 2016).

[Figure]

Volume fraction remaining (VFR)as a function of heating temperature for 8 samples

[Figure]

A map showing the location of the SORPES site (Ding et al., 2013). The prevailing wind was

from northeast during winter, and from southeast during summer.

Ding, A. J., Fu, C. B., Yang, X. Q., Sun, J. N., Zheng, L. F., Xie, Y. N., Herrmann, E., Nie, W., Petäjä, T., Kerminen, V. M., and Kulmala, M.: Ozone and fine particle in the western Yangtze River Delta: an overview of 1 yr data at the SORPES station, Atmos. Chem. Phys., 13, 5813-5830, 10.5194/acp-13-5813-2013, 2013.

Ding, A., Nie, W., Huang, X., Chi, X., Sun, J., Kerminen, V.-M., Xu, Z., Guo, W., Petäjä, T., Yang, X., Kulmala, M., and Fu, C.: Long-term observation of air pollution-weather/climate interactions at the SORPES station: a review and outlook, Front. Environ. Sci. Eng., 10, 15, 10.1007/s11783-016-0877-3, 2016.

**B) Analytical procedure for the preparation of HULIS extracts**

The authors blindly believe that a whole fraction of HULIS is greatly soluble in the aqueous phase. However, in the light of recent papers dealing with the chemical composition HULIS (e.g., *Environ. Sci. Technol.*, 2016, 50 (4), 1721; *Atmos. Chem.*, 2015, 72, 65), aerosol-derived HUmic LIke Substances represent a complex chemical mixtures, including high-molecular-weight aliphatics (primarily C27–C32) with small proportions of −CH3, −OH, and C═O groups. These are poorly soluble in water, thus the extraction with pure water only may lead to the substantial loss of analyte to be further subjected to the H-TDMA analysis. I am highly surprised that the authors did not take it into account despite it is a basic approach in the analytical atmospheric chemistry: a sample preparation is the most crucial factor. I would suggest broadening a result discussion with providing additional data obtained for samples extracted with less polar solvents, say acetonitrile and water acetonitrile (50/50v). The same could be applied for another solvent couple: methanol and methanol-water (50/50), as recently have been suggested by Lin (*Environ. Sci. Technol.*, 2014, 48(20), 12012). Moreover, I am a bit concern of the selection of SPE method for HULIS water extracts since it may result in a dramatic loss of highly oxidized and water soluble products, such as organosulfates (nitroxyorganosulfates).

Response: Thanks for the comment.

It is an issue of definition about HULIS. HULIS is operationally defined by the procedure that is used for its isolation. HULIS used in this work refers to water-soluble part of humic-like substance (or the part of water-soluble organic compounds that are hydrophobic). The referee's definition of HULIS is broader, including both the water-soluble and water-insoluble parts. We will add some statements in the revised manuscript.

Actually, the referee provided a good suggestion. Volatility tests of HULIS with a broader definition (both water-soluble and water-insoluble parts) have been planned in the further work.

---

## Author Comment (AC2) · 8 Feb 2017

Referee #4

General Comments

Here the authors report results of a laboratory study of the volatility of HULIS extracted from aerosol samples collected at a rural site in eastern China. Samples were atomized, four different sizes were selected with a DMA, and then aerosol was passes through a thermal denuder to measure changes in size with increasing temperature. Extracts were also mixed with ammonium sulfate prior to atomization to investigate the effects of salt-organic interactions on volatility. The volatility profiles were analyzed using a model in which various parameters (heat of vaporization, molecular weight, etc.) were assigned based on previous studies and the aerosol was distributed among three volatility bins (SVOC, LVOC, ELVOC) using the model-measurement comparison. The results of AMS measurements indicate that HULIS is highly oxidized (O/C _ 1 or greater) and the volatility measurements show that most of the HULIS is low and extremely low volatility material, consistent with the high degree of oxidation. Small decreases in volatility were also observed when ammonium sulfate was added that indicate chemical interactions between the organic and inorganic materials. The explanations for the general trends observed in the data and model results are reasonable, and overall there are no real surprises. This is a pretty straightforward study, the experimental and modeling components are well done, and the data interpretation is reasonable. The paper is a useful contribution to the literature and is worthy of publication in ACP. I have only a few minor comments that should be addressed.

Specific Comments

1. Line 194–197: What is the fraction of HULIS in the organic component of the samples?

Response: The HULIS-C made about 30% of the total organic carbon (OC). We will add this information into the revised manuscript.

2. Line 199–201: Is auto-oxidation a potential source for these HOMs?

Response: Yes, it is possible.

Aromatics have been demonstrated to form HOMs via auto-oxidation, which would be one possible source of HULIS (Molteni et al., 2016).

We will add some discussion on this into the revised manuscript.

*Molteni, U., Bianchi, F., Klein, F., El Haddad, I., Frege, C., Rossi, M. J., Dommen, J., and Baltensperger, U.: Formation of highly oxygenated organic molecules from aromatic compounds, Atmos. Chem. Phys. Discuss., 2016, 1-39, 10.5194/acp-2016-1126, 2016.*

3. What are the effects of assumed model parameters on the interpretation of experimental results? Were sensitivity studies conducted? For example, there is ongoing debate about the appropriate value of the mass accommodation coefficient, which may range from about 1 to 0.001. Couldn't changes in these parameters with organic and inorganic composition be responsible for the observations rather than changes in the SVOC, LVOC, and ELVOC fractions? Some discussion of these issues is needed.

Response: Thanks for the comment.

This is actually a good question. The value of mass accommodation coefficient (MAC) did influence the simulated distribution of SVOC, LVOC and ELVOC. What we need is to choose a MCA value that the model can best reproduce the measured evaporation behavior. As showed in the following figures, sensitivity of the kinetic evaporation model was tested towards different MAC values (i.e. MAC=1, 0.1, 0.01) for both pure HULIS sample and mixed samples. It was obviously that only when MAC was set to 1, the simulated thermogram showed the best agreement with the observation. This is the reason we chose 1 as the MAC value in the MS.

We will add some discussion on this in the revised manuscript.

[Figure]

[Figure]

Comparison of measured VFR with modeled VFR for HULIS of sample 1, with accommodation coefficient of 1 in left panel, 0.1 middle panel, and 0.01 in right panel.

[Figure]

[Figure]

Comparison of measured VFR with modeled VFR for 1:1 mixed sample of HULIS and AS, with accommodation coefficient of 1 in left panel, 0.1 middle panel, and 0.01 in right panel.

Technical Comments

Line 61: "Abortion" should be "absorption".

Response: Thanks. We will correct it in the revised manuscript.

Line 128: "An- alyzer" should be "Analyzer".

Response: Thanks. We will correct it in the revised manuscript.

---

## Author Comment (AC3) · 8 Feb 2017

Referee #1

This manuscript presents the first study to my knowledge of the volatility of humiclike substances (HULIS) in atmospheric aerosols. The results ultimately link HULIS, which has been observed ubiquitously in atmospheric aerosols using filter collection and extraction, with low and extremely low volatility organic material (ELVOC), which has been observed to be similarly common using AMS and thermal denuder-type techniques. The authors also demonstrate that interactions between HULIS and inorganic salts such as sulfate can greatly decrease the already low volatility of the HULIS. This is a very significant contribution to the field, and should be published after a few issues are addressed.

- Another reviewer commented on the relatively few filter samples that were used in this study. Can the authors make an argument to justify this, for example, were the properties of the HULIS on these filters representative of other samples taken at the same location?

Response: Thanks for the comment and suggestion.

Please refer to the response of comment A of referee 3. We actually analyzed 8 HULIS samples, which were collected during both winter and summer. The volatility of all these 8 samples behaved in quite a similar way. We thus selected 4 samples (the PM concentrations of 2 of them were high, and the other 2 were low) to make the arguments clearer.

- My main concern is the extensive chemical processing of the HULIS during isolation before analysis. Is there a way for us to know how this may have impacted the volatility or other properties of the material as compared to the real state in ambient aerosols? A discussion of this point in the manuscript is needed.

Response: Thanks for the comment.

We agree the isolation processes would influence the properties of HULIS as compared to those in real ambient aerosols, especially when inorganic ions were mostly removed. In this work, the sampling site was in Yangtze River Delta, the aerosol of which was dominated by inorganic salts, especially ammonium sulfate which accounted for about 30%

of PM$_{2.5}$ (Xie et al., 2015). This was one of the reason to investigate the possible interaction between ammonium sulfate and HULIS in the manuscript. We will add some discussion on this into the revised manuscript.

What we want to emphasize here is that the volatility (as well as some other physical properties like hygroscopicity) is an overall property/nature of an aerosol that is related not only to the volatility of each individual compounds but also to interactions between these compounds. In case of thousands of compounds in a real atmospheric aerosol, its volatility, especially for organic aerosol, behaves in a very complicated way. Currently, volatility studies on OA have mostly focused on laboratory-generated organic particles or ambient particles. Laboratory-generated organic particles are very far from the real ambient particles, whereas ambient particles are too complex to be understood. Therefore, as we stated in the introduction, one possible way is to isolate some classes of organic compounds of the aerosol and analyze their volatility separately. Here, in this work, the volatility analysis of extracted HULIS from real ambient aerosol was one of such attempts.

- Related to my last point, since the HULIS extraction process removes inorganic ions from the sample, are we to understand that the samples in which salts have been re-introduced are more representative of the true volatility of HULIS in atmospheric aerosols? If so, this should be emphasized.

Response: Thanks for the comment.

We agree with referee's viewpoint and will add some discussion into the revised manuscript.

- Some language editing is necessary in places, for example, lines 42-43

Response: Thanks. We will re-edit the language in the revised manuscript.

---

## Author Response (AR2)

Referee #1

This manuscript presents the first study to my knowledge of the volatility of humiclike substances (HULIS) in atmospheric aerosols. The results ultimately link HULIS, which has been observed ubiquitously in atmospheric aerosols using filter collection and extraction, with low and extremely low volatility organic material (ELVOC), which has been observed to be similarly common using AMS and thermal denuder-type techniques. The authors also demonstrate that interactions between HULIS and inorganic salts such as sulfate can greatly decrease the already low volatility of the HULIS. This is a very significant contribution to the field, and should be published after a few issues are addressed.

- Another reviewer commented on the relatively few filter samples that were used in this study. Can the authors make an argument to justify this, for example, were the properties of the HULIS on these filters representative of other samples taken at the same location?

Response: Thanks for the comment and suggestion.

Please refer to the response of comment A of referee 3. We actually analyzed 8 HULIS samples, which were collected during both winter and summer. The volatility of all these 8 samples behaved in quite a similar way. We thus selected 4 samples (the PM concentrations of 2 of them were high, and the other 2 were low) to make the arguments clearer. We have added the following discussion in the revised manuscript.

Line 151-156: "In this work, we totally analyzer 8 samples collected during both winter and summer, and covering a wide range of PM concentration from less than 40 $\mu g/m^3$ to higher than 150 $\mu g/m^3$. All these 8 samples showed similar evaporation behavior with some small differences in details (figures not shown). Therefore, in terms of volatility, we believe that there were no large differences between the collected HULIS samples. We finally selected 4 samples with different PM concentrations to represent the HULIS samples at the SORPES station and made the argument clear."

- My main concern is the extensive chemical processing of the HULIS during isolation before analysis. Is there a way for us to know how this may have impacted the volatility

or other properties of the material as compared to the real state in ambient aerosols? A discussion of this point in the manuscript is needed.

Response: Thanks for the comment.

We agree the isolation processes would influence the properties of HULIS as compared to those in real ambient aerosols, especially when inorganic ions were mostly removed. In this work, the sampling site was in Yangtze River Delta, the aerosol of which was dominated by inorganic salts, especially ammonium sulfate which accounted for about 30% of $PM_{2.5}$ (Xie et al., 2015). This was one of the reason to investigate the possible interaction between ammonium sulfate and HULIS in the manuscript. We have added the following discussions, which was highlighted with blue in the revised manuscript.

*Line 127-131: "In case that the isolation processes may influence the evaporation behavior of HULIS by removing some species (especially the inorganic salts) which were originally mixed together with HULIS, we also re-induce ammonium sulfate, the most important inorganic salt, to the extracted HULIS and investigate the volatility of the mixed samples (section 3.2)."*

*Line 276: "Organic matters, including HULIS, are always mixed with inorganic species in the real ambient aerosol."*

*Line 324-327: "It should be emphasized here in case HULIS are always mixed with ammonium sulfate, which accounted for 30% of the mass of $PM_{2.5}$ (Xie et al., 2015), in ambient aerosols of YRD region, it is possible that these mixed samples are more representative of the real volatility of HULIS in ambient aerosols."*

What we want to emphasize here is that the volatility (as well as some other physical properties like hygroscopicity) is an overall property/nature of an aerosol that is related not only to the volatility of each individual compounds but also to interactions between these compounds. In case of thousands of compounds in a real atmospheric aerosol, its volatility, especially for organic aerosol, behaves in a very complicated way. Currently, volatility studies on OA have mostly focused on laboratory-generated organic particles or ambient particles. Laboratory-generated organic particles are very far from the real

ambient particles, whereas ambient particles are too complex to be understood. Therefore, as we stated in the introduction, one possible way is to isolate some classes of organic compounds of the aerosol and analyze their volatility separately. Here, in this work, the volatility analysis of extracted HULIS from real ambient aerosol was one of such attempts.

- Related to my last point, since the HULIS extraction process removes inorganic ions from the sample, are we to understand that the samples in which salts have been re-introduced are more representative of the true volatility of HULIS in atmospheric aerosols? If so, this should be emphasized.

Response: Thanks for the comment.

We agree with referee's viewpoint. Please refer to the response of last comment. We have added some discussion into the revised manuscript.

- Some language editing is necessary in places, for example, lines 42-43

Response: Thanks. We have re-edited the language in the revised manuscript.

E.g. Line 41-42: *"suggesting that the interaction with ammonium sulfate tends to decrease the volatility of atmospheric organic compounds."*

Referee 3

In my opinion the submitted manuscript addresses an interesting scientific issue, i.e., the study on the volatility of HULIS aerosol compounds. The text is understandable for the reader. The objectives are clearly defined and raised conclusions are coherent. However, I would opt to likely consider moving the manuscript first to the ACPD full review process. This is because I have a few remarks as stated below:

**A) Limited number of ambient samples**

The authors made their research based only on 4 filter samples. No justification was

provided as to the choice of such a limited number of samples. Certainly, during the full ACPD review phase the authors would be requested to build up their scientific story on the results obtained from the analysis of more ambient samples, say 10 or larger due to the increased complexity of HULIS (e.g., *Environ. Sci. Technol.*, 2016, 50 (4), 1721).

Response: Thanks for the comment, which mainly focused on the representative of our samples.

Firstly, we agree that HULIS are a very complex mixture that contains a large number of compounds with different molecular structures. However, based on our results, their overall volatility of different HULIS samples behaved in quite a similar way (which is reasonable as volatility is mostly controlled by molecule weight and oxidation state). In this work, we actually analyzed 8 samples collected during both winter and summer, and covering a wide range of PM concentration from less than 40 µg/m$^3$ to higher than 150 µg/m$^3$. All these 8 samples showed similar evaporation behavior (as showed in the following figures), with some small differences in details. Therefore, in terms of volatility, we believe that there were no large differences between the collected HULIS samples. We finally selected 4 samples with different PM concentrations to represent the HULIS samples at the SORPES station and made the argument simple and clear that ambient HULIS overall showed a low volatility. We have added the following discussion in the revised manuscript.

*Line 151-156: "In this work, we totally analyzer 8 samples collected during both winter and summer, and covering a wide range of PM concentration from less than 40 µg/m$^3$ to higher than 150 µg/m$^3$. All these 8 samples showed similar evaporation behavior with some small differences in details (figures not shown). Therefore, in terms of volatility, we believe that there were no large differences between the collected HULIS samples. We finally selected 4 samples with different PM concentrations to represent the HULIS samples at the SORPES station and made the argument clear."*

Secondly, our sampling site, named SORPES, has been demonstrated to be a regional background site of Yangtze River Delta. As showed in the followed 2$^{nd}$ figure (Ding et al.,

2013), SORPES station located in the downwind region of YRD, and is about 20 km east of the downtown Najing city. Compared to the typical urban measurement station, air masses arriving at this site are more aged and well-mixed. Secondary formed aerosol, e.g. sulfate, nitrate and SOA, was dominated the mass of $PM_{2.5}$. The samples collected at this site were thus believed to be regional representation of Yangtze River Delta (Ding et al., 2013 & 2016). We have added the following sentence in the revised manuscript.

*Line 109-110: "The samples collected here, especially for the regional polluted days, were believed to be regional representation of YRD."*

[Figure]

Volume fraction remaining (VFR)as a function of heating temperature for 8 samples

[Figure]

A map showing the location of the SORPES site (Ding et al., 2013). The prevailing wind was from northeast during winter, and from southeast during summer.

Ding, A. J., Fu, C. B., Yang, X. Q., Sun, J. N., Zheng, L. F., Xie, Y. N., Herrmann, E., Nie, W., Petäjä, T., Kerminen, V. M., and Kulmala, M.: Ozone and fine particle in the western Yangtze River Delta: an overview of 1 yr data at the SORPES station, Atmos. Chem. Phys., 13, 5813-5830, 10.5194/acp-13-5813-2013, 2013.

Ding, A., Nie, W., Huang, X., Chi, X., Sun, J., Kerminen, V.-M., Xu, Z., Guo, W., Petäjä, T., Yang, X., Kulmala, M., and Fu, C.: Long-term observation of air pollution-weather/climate interactions at the SORPES station: a review and outlook, Front. Environ. Sci. Eng., 10, 15, 10.1007/s11783-016-0877-3, 2016.

**B) Analytical procedure for the preparation of HULIS extracts**

The authors blindly believe that a whole fraction of HULIS is greatly soluble in the aqueous phase. However, in the light of recent papers dealing with the chemical composition HULIS (e.g., *Environ. Sci. Technol*., 2016, 50 (4), 1721; *Atmos. Chem.*, 2015, 72, 65), aerosol-derived HUmic LIke Substances represent a complex chemical mixtures, including high-molecular-weight aliphatics (primarily C27–C32) with small proportions of −CH3, −OH, and C═O groups. These are poorly soluble in water, thus the extraction with pure water only may lead to the substantial loss of analyte to be further subjected to the H-TDMA analysis. I am highly surprised that the authors did not take it into account despite it is a basic approach in the analytical atmospheric chemistry: a sample preparation is the most crucial factor. I would suggest broadening a result discussion with providing additional data obtained for samples extracted with less polar solvents, say acetonitrile and water acetonitrile (50/50v). The same could be applied for

another solvent couple: methanol and methanol-water (50/50), as recently have been suggested by Lin (*Environ. Sci. Technol*., 2014, 48(20), 12012). Moreover, I am a bit concern of the selection of SPE method for HULIS water extracts since it may result in a dramatic loss of highly oxidized and water soluble products, such as organosulfates (nitroxyorganosulfates).

Response: Thanks for the comment.

It is an issue of definition about HULIS. HULIS is operationally defined by the procedure that is used for its isolation. HULIS used in this work refers to water-soluble part of humic-like substance (or the part of water-soluble organic compounds that are hydrophobic). The referee's definition of HULIS is broader, including both the water-soluble and water-insoluble parts. We have added the following sentence in the revised manuscript to make this point clearer.

Line 126-127: "It should be noted here that HULIS extracted in this work refers to the part of water-soluble organic compounds that are hydrophobic."

Actually, the referee provided a good suggestion. Volatility tests of HULIS with a broader definition (both water-soluble and water-insoluble parts) have been planned in the further work.

Referee #4

General Comments

Here the authors report results of a laboratory study of the volatility of HULIS extracted from aerosol samples collected at a rural site in eastern China. Samples were atomized, four different sizes were selected with a DMA, and then aerosol was passes through a thermal denuder to measure changes in size with increasing temperature. Extracts were also mixed with ammonium sulfate prior to atomization to investigate the effects of salt-organic interactions on volatility. The volatility profiles were analyzed using a model in which various parameters (heat of vaporization, molecular weight, etc.) were assigned

based on previous studies and the aerosol was distributed among three volatility bins (SVOC, LVOC, ELVOC) using the model-measurement comparison. The results of AMS measurements indicate that HULIS is highly oxidized (O/C _ 1 or greater) and the volatility measurements show that most of the HULIS is low and extremely low volatility material, consistent with the high degree of oxidation. Small decreases in volatility were also observed when ammonium sulfate was added that indicate chemical interactions between the organic and inorganic materials. The explanations for the general trends observed in the data and model results are reasonable, and overall there are no real surprises. This is a pretty straightforward study, the experimental and modeling components are well done, and the data interpretation is reasonable. The paper is a useful contribution to the literature and is worthy of publication in ACP. I have only a few minor comments that should be addressed.

Specific Comments

1. Line 194–197: What is the fraction of HULIS in the organic component of the samples?

Response: The HULIS-C made about 30% of the total organic carbon (OC). We have added the following information into the revised manuscript.

*Line 212-214: "The HULIS concentrations were also higher in samples 1 and 2 (about 9 µg/m³, ratio of HULIS-carbon to OC were about 0.3) than in samples 3 and 4 (about 6 µg/m³, ratio of HULIS-carbon to OC were about 0.4)."*

2. Line 199–201: Is auto-oxidation a potential source for these HOMs?

Response: Yes, it is possible.

Aromatics have been demonstrated to form HOMs via auto-oxidation, which would be one possible source of HULIS (Molteni et al., 2016).

We have added the following discussion and reference into the revised manuscript.

*Line 219-221: "One possible source of these HOMs is the oxidation of aromatics, which initiated by hydroxyl radical (OH) and followed by auto-oxidation (Molteni et al., 2016)."*

*Molteni, U., Bianchi, F., Klein, F., El Haddad, I., Frege, C., Rossi, M. J., Dommen, J., and Baltensperger, U.: Formation of highly oxygenated organic molecules from aromatic compounds, Atmos. Chem. Phys. Discuss., 2016, 1-39, 10.5194/acp-2016-1126, 2016.*

3. What are the effects of assumed model parameters on the interpretation of experimental results? Were sensitivity studies conducted? For example, there is ongoing debate about the appropriate value of the mass accommodation coefficient, which may range from about 1 to 0.001. Couldn't changes in these parameters with organic and inorganic composition be responsible for the observations rather than changes in the SVOC, LVOC, and ELVOC fractions? Some discussion of these issues is needed.

Response: Thanks for the comment.

This is actually a good question. The value of mass accommodation coefficient (MAC) did influence the simulated distribution of SVOC, LVOC and ELVOC. What we need is to choose a MCA value that the model can best reproduce the measured evaporation behavior. As showed in the following figures, sensitivity of the kinetic evaporation model was tested towards different MAC values (i.e. MAC=1, 0.1, 0.01) for both pure HULIS sample and mixed samples. It was obviously that only when MAC was set to 1, the simulated thermogram showed the best agreement with the observation. This is the reason we chose 1 as the MAC value in the MS.

We have added the following discussion on this in the revised manuscript.

*Line 181-185: "Since the value of mass accommodation coefficient (MAC) may influence the simulated volatility distribution of HULIS, sensitivity of this kinetic evaporation model was tested towards different values of mass accommodation coefficient (i.e. MAC=1, 0.1, 0.01) for both pure HULIS sample and mixed samples (figure not shown). The results suggested that 1 is the proper MAC value to best reproduce the measured evaporation behavior (Table 1)."*

[Figure]

Comparison of measured VFR with modeled VFR for HULIS of sample 1, with accommodation coefficient of 1 in left panel, 0.1 middle panel, and 0.01 in right panel.

[Figure]

Comparison of measured VFR with modeled VFR for 1:1 mixed sample of HULIS and AS, with accommodation coefficient of 1 in left panel, 0.1 middle panel, and 0.01 in right panel.

Technical Comments

Line 61: "Abortion" should be "absorption".

Response: Thanks. We have corrected it in the revised manuscript.

Line 128: "An- alyzer" should be "Analyzer".

Response: Thanks. We have corrected it in the revised manuscript.

---

## Author Response (AR3)

Response to the comments of Dr. Jason Surratt:

1) Extractions and HULIS preparations:

Like Reviewer 2 and 3 stated, I still have concerns with acidifying the water extracts before SPE separation. The reason for this concern is if the acidity of the extract is too high in the presence of water you will likely induce acid-catalyzed hydrolysis of certain compounds such as hydroperoxides or even catalyze further reactions that produce lower-volatility OA than originally present in the original aerosol sample. Shouldn't the authors more clearly state that based on the current set of measurements that this can't be ruled out? Basically, at least directly acknowledge the limitations of the current work. I feel this limitation is weakly stressed in the revised text.

Response: Thanks for the comment.

The main reason for acidifying the water extracts is to make the organic poly-acids in the neutral form rather than ionic form so that they can be retained by the SPE sorbent. We acidified the water extract right before loading the sample on the SPE cartridge. That is a very short period (less than 15 min, generally). Since this is in diluted water solution rather than condensed-phase, I think the acid-catalyzed reactions won't take place significantly at this time scale (e.g. Birdsal et al., 2013). We have added this information in the revised manuscript as following.

Line 122-125: "Since the HULIS samples were in diluted water solutions rather than condensed-phase, and were acidified right before (in general, less than 15 min) loading the sample on the SPE cartridge, we believe the acid-catalyzed reactions unlikely would take place to a significant degree as to influence their volatility (e.g. Birdsall et al., 2013)."

Birdsall, A. W., Zentner, C. A., and Elrod, M. J.: Study of the kinetics and equilibria of the oligomerization reactions of 2-methylglyceric acid, Atmos. Chem. Phys., 13, 3097-3109, 10.5194/acp-13-3097-2013, 2013.

Related to this, the authors should be aware that I think the statement lines 120-122 "It should be noted here that HULIS extracted in this work refers to part of the WSOCs that

are hydropobic" is unlcear. Should one really say ".....that are partly hydrophobic in character." They wouldn't be in the water extract if they were completely hydrophobic, right? I can see potential readers of your article being confused by this wording.

Response: Agree, and have changed in the revised manuscript.

Since the extraction is very specific to water, for wording like on line 155 "HULIS samples," I would be clearer here and throughout the remainder of the text that this is the "water-soluble HULIS fraction." Please change this so readers don't get confused that you mean the entire fraction of HULIS. I agree with reviewer 3's concerns on the extraction issue.

Response: Agree, and have changed in the revised manuscript.

2)      One point I want to make after re-reading this manuscript. The authors assume that ELVOCs explain the volatility behavior of the water-soluble fraction of HULIS discussed throughout the manuscript. However, it is unclear to me what chemical data exists at the molecular level to support this statement. The water-soluble fraction can include may compounds resulting from heterogeneous reactions (e.g., acid-catalyzed ring-opening reactions of epoxides, like IEPOX), or other processes. I suggest that more care be taken here. Can the authors at least add a statement clarifying this. All of the pathways you describe could tentatively explain these compounds and I don't think the chemical data are strong enough to emphasize one over any others. As of now, all are tentative proposals.

Response: Thanks for the comment.

We agree that we did not have enough molecular information for the extracted HULIS to support their volatility estimation. But in this work, our target was to evaluate the volatility of water-soluble HULIS more directly based on the VTDMA measurement, which was believed to be more reliable. One of our previous studies at another city clusters of Pearl River Delta (PRD) in east China have provided detailed molecular information of extracted HULIS (Lin et al., 2012). In that study, the detected molecular weight was in the range from 100 to 500 g/mol, and contained compounds of CHON,

CHO, CHOS and CHONS. With the method provided by Li et al. (2016) to estimate the volatility of these compounds, we can confirm that a large fraction of these compounds was LVOC or ELVOC, especially for the S-containing compounds with molecular weight higher than 200. We will investigate the molecular information of the extracted HULIS in the future study, and connect the estimated volatility to the measured results. We added the following sentences in the revised manuscript.

*Line 252-257: "Detailed molecular information of extracted HULIS was not available in this study. But a previous study at the Pearl River Delta, another polluted megacity region in China, showed the molecular weight of HULIS was in the range of 100 to 500, with a significant fraction higher than 260. Using the method provided by Li et al. (2016) to estimate the volatility of these compounds, we calculated that a large fraction of these compounds in water-soluble HULIS was LVOC or ELVOC, especially the S-containing compounds with molecular weight higher than 200. Future volatility measurement studies are suggested to investigate the S-containing compounds."*

For the issue of potential acid-catalyzed reactions, please refer to our response to comment 1. We think the acid-catalyzed reactions won't take place significantly in a time scale less than 15 minutes.

*Lin, P., Rincon, A. G., Kalberer, M., and Yu, J. Z.: Elemental Composition of HULIS in the Pearl River Delta Region, China: Results Inferred from Positive and Negative Electrospray High Resolution Mass Spectrometric Data, Environ Sci Technol, 46, 7454-7462, 10.1021/es300285d, 2012.*

*Li, Y., Pöschl, U., and Shiraiwa, M.: Molecular corridors and parameterizations of volatility in the chemical evolution of organic aerosols, Atmos. Chem. Phys., 16, 3327-3344, 10.5194/acp-16-3327-2016, 2016.*

3)      Revised line 151 stating "we totally analyzer 8 samples":  Dont' the authors mean to say: "we analyzed a total of 8 samples collected during both winter and summer," or do they mean to say 4 total in summer and 4 total in winter? This is unclear from the revised text added to the manuscript.

Response: Agree, and have changed to the following sentence in the revised manuscript.

*Line 154-155: "In this work, we totally analyzer 8 samples with 6 of them collected during winter and the other two during summer."*

4)      Revised Lines 155-156: This is a poorly worded sentence. Can the authors revise this to something like: "Of the 8 samples analyzed, we selected only 4 of the samples representing different PM concentrations to represent the ranging types of HULIS samples collected at the SORPES station." The last part of this sentence "and made the argument clear..." seems to be an incomplete sentence. Please correct this.

Response: Agree and have changed the statement in the revised manuscript.

*Line158-160: "Of the 8 samples analyzed, we therefore selected only 4 of the samples representing different PM concentrations to represent the ranging types of HULIS samples collected at the SORPES station."*

5)  Revised Line 276: Change "Organic matters, ..." to "Organic material, ...."

Response: Agree and have changed in Line 286 in revised manuscript.

6)      Revised Line 153 stating "(figures not shown): Can't the authors add the figures to the SI section? I think interested readers may want to see these details.

Response: Agree and have added the figure to the SI section.